# Comparing Rubber Bearings and Eradi-Quake System for Seismic Isolation of Bridges

**DOI:** 10.3390/ma13225247

**Published:** 2020-11-20

**Authors:** Chang Beck Cho, Young Jin Kim, Won Jong Chin, Jin-Young Lee

**Affiliations:** 1Department of Infrastructure Safety Research, Korea Institute of Civil Engineering and Building Technology, Goyang 10223, Korea; cbcho@kict.re.kr (C.B.C.); yjkim@kict.re.kr (Y.J.K.); wjchin@kict.re.kr (W.J.C.); 2School of Agricultural Civil & Bio-industrial Engineering, Kyungpook National University, Daegu 41566, Korea

**Keywords:** seismic isolator, seismic isolation of bridges, natural rubber bearing, lead rubber bearing, Eradi-Quake System (EQS), energy dissipated per cycle (EDC)

## Abstract

Seismic isolation systems have been used worldwide in bridge structures to reduce vibration and avoid collapse. The seismic isolator, damper, and Shock Transmission Unit (SUT) are generally adopted in the seismic design of bridges to improve their seismic safety with economic efficiency. There are several seismic isolation systems, such as Natural Rubber Bearing (NRB), Lead Rubber Bearing (LRB), and the Eradi-Quake System (EQS). EQS as a new technology is expected to effectively reduce both seismic force and displacement, but there is still some need to verify whether it might provide an economical and practical strategy for a bridge isolation system. Moreover, it is important to guarantee consistent performance of the isolators by quality control. A comparative evaluation of the basic properties of the available seismic isolators is thus necessary to achieve a balance between cost-effectiveness and the desired performance of the bridge subjected to extreme loading. Accordingly, in this study, the seismic response characteristics of the seismic isolation systems for bridges were investigated by conducting compressive test and compressive-shear test on NRB, LRB, and EQS.

## 1. Introduction

### 1.1. General

Seismic isolation systems such as the seismic isolator, damper, and Shock Transmission Unit (SUT) have been widely used in bridge structures to reduce vibration, prevent collapse, and improve seismic safety with economic efficiency. Among these systems, the seismic isolation bearing or seismic isolator allows the structure to behave more flexibly with enhanced seismic response by decoupling the superstructure from the foundation and providing additional damping. The isolators are laterally flexible elements but can carry the vertical loads of the overlying structure. Since the isolators are more flexible than the structure, most of the lateral movements caused by the earthquake occur in the isolators which make the isolated structure experience less motion and reduced forces. Compared to a traditional non-isolated structure, seismic isolation can reduce the forces and displacements in the structure by up to 75% that allow the structure to remain functional with less or no damage. The cost saving brought by the seismic isolation is achieved by the possibility to reduce the dimensions of the foundation by dropping the elastic forces down by up to 75% in the design [1,2,3,4].

There are several seismic isolation systems such as Natural Rubber Bearing (NRB), Lead Rubber Bearing (LRB), and Eradi-Quake System (EQS). EQS as a new technology has been proven to effectively reduce both seismic force and displacement [5,6,7,8], but there is still the need to verify whether it might provide an economical and practical strategy for bridge isolation systems. Moreover, it is important to guarantee consistent performance of the isolators by quality control. A comparative evaluation of the basic properties of the available seismic isolators is thus necessary to achieve a balance between cost-effectiveness and the desired performance of the bridge subjected to extreme loading. Accordingly, in this study, the seismic response characteristics of the seismic isolation systems are investigated by conducting a compressive test and compressive-shear test on NRB, LRB, and EQS.

### 1.2. Elastomeric Bearings

The concept of base isolation is quite simple. The system decouples the structure from the horizontal components of the ground motion by interposing structural elements with low horizontal stiffness between the structure and the foundation [9]. The first use of a rubber isolation system to protect a structure from earthquakes happened in 1969 for an elementary school in Skopje, Yugoslavia [10]. Based on their main properties and composition, there are two types of elastomeric bearings: NRB (Figure 1a) and LRB (Figure 1b).

NRBs are made of alternating steel shims vulcanized or glued together and elastomeric layers which provide lateral flexibility and elastic restoring force. The steel plates reinforce the bearing by providing vertical load capacity and preventing lateral bulge. A rubber cover protects the ensemble. Mounting plates connect the device to the structure above and below. Depending on the elastomeric compounds used, NRBs are available as either low damping or high damping. Low damping NRBs are used in conjunction with supplementary damping devices, whereas high damping NRBs can be used without other devices since they provide sufficient inherent damping [3,9,11,12,13,14]. LRBs are similar to NRBs but contain a lead core. The steel shims confine the lead plug and force it to deform in shear. Along with this deformation, dissipation of energy takes place.

### 1.3. EQS

EQS is an innovative sliding-type seismic isolation bearing system designed to minimize forces and displacements experienced by structures subjected to seismic load. As shown in Figure 2, its components consist of top and bottom plates which are fixed to the superstructure and substructure of the bridge, respectively. In the bottom plate, a Poly Tetra Fluoro Ethylene (PTFE) friction disc, which has a primary stiffness, performs friction damping and dissipates seismic energy with resistant force caused by its sliding resilient friction. The lateral Mass Energy Regulator (MER) springs that supply restoring forces primarily reposition the plate back to its original position after finishing displacement responses. EQS transfers the seismic energy into heat and spring energy through friction of a PTFE disc pad and MER [2,6,7,8,15,16,17,18,19,20,21]. Whereas the rubber bearings exhibit relatively significant dispersion in their mechanical parameters, EQS can be tuned to the desired level of stiffness and damping, and generally has a force–displacement relationship presenting post-yield stiffness-hardening during large inelastic displacement.

## 2. Experimental Program

### 2.1. Design Properties of Test Specimens

The seismic resistance of NRB, LRB, and EQS was evaluated experimentally on a total of seven specimens (NRB: 3 specimens, LRB: 3 specimens, EQS: 1 specimen). Table 1 lists the properties of the NRB and LRB specimens where *d* is the diameter of rubber; *P_v_* is the vertical load bearing capacity; *G* is the shear modulus; *K_v_* and *K_h_* are the vertical and horizontal stiffnesses, respectively; *P_y_* is the yield strength; EDC is the energy dissipated per cycle, which is the energy dissipation capacity of the isolator when it displays a reciprocating motion under seismic load; and *P_b_* is the buckling load. For further comparison, the NRB and LRB specimens were designed to sustain the same *P_v_* of 2945 kN and have the same diameter and shear modulus. Table 2 arranges the design properties of the EQS specimens designed for *P_v_* = 2775 kN and design maximum displacement (*d*_max_) of 50 mm. In Table 2, *L_dead_*, *L_live_*, and *L* are the design dead load, the design live load, and design load, respectively. In addition, *f_max_* is the coefficient of friction of the PTFE disc at high velocity of sliding, and *f_min_* is the coefficient of friction at essentially zero velocity of sliding. *K_eff_* denotes the effective stiffness.

### 2.2. Testing Method and Setup

#### 2.2.1. Compressive Load Test

The setup adopted for the compressive test is illustrated in Figure 3. Biaxial loading was applied using two hydraulic actuators. The vertical actuator (Samyeon Tech, Daegu, Korea) with a capacity of 3500 kN and the horizontal actuator (Samyeon Tech, Daegu, Korea) with a horizontal capacity of 500 kN are components of the testing machine and can be extended up to 150 and 600 mm, respectively. A Linear Variable Displacement Transducer (LVDT) (Samyeon Tech, Daegu, Korea) was installed to measure the biaxial displacements, as depicted in Figure 3c for the NRB and LRB specimens and Figure 3d for the EQS specimens. The loading details are explained in Figure 4 and Table 3. The design vertical load (*p*_0_) was 2000 kN for LRB and NRB, and 1850 kN for EQS. The compressive loads were imparted three times on the specimens within a range between *p*_1_ and *p*_2_, which are 70% and 130% of the design vertical load. The compressive stiffness, *K_v_*, can be calculated by Equation (1) with the test data, *p*_1_, *p*_2_, *Y*_1_, and *Y*_2_, where *Y*_1_ and *Y*_2_ are the horizontal displacements corresponding to *p*_1_ and *p*_2_, respectively, as shown in Figure 4. Moreover, the compressive stresses, $f_{1}$ and $f_{2,}$ can be calculated by Equations (2) and (3) using the compressive loads, *p*_1_ and *p*_2_.
(1)$$K_{v}=\frac{p_{2}-p_{1}}{Y_{2}-Y_{1}},$$
(2)$$p_{1}=A_{\mathrm{load}}\cdot f_{1},$$
(3)$$p_{2}=A_{\mathrm{load}}\cdot f_{2},$$
where $A_{\mathrm{load}}$ is the net loading area excluding the area of the lead core in the LRB specimens.

#### 2.2.2. Compressive-Shear Test

The specimens were tested according to the test guidelines for bridge isolators [22] developed by the Korea Institute of Civil Engineering and Building Technology (KICT) using a compressive-shear testing machine (Samyeon Tech, Daegu, Korea), the conceptual drawing of which is shown Figure 5. The shear force and the horizontal displacement were measured under constant vertical load. During the testing, the loading plates should be parallel to each other and the actuator must apply a constant vertical load when the height of the specimen is changed. The lateral (shear) loading history is shown in Figure 6. The shear stiffness and the equivalent damping can be calculated using the measured test results. The shear force measured by a horizontally installed load cell had to be calibrated after the test because the friction force of the testing machine was added to the measured load, even though the bearings were installed to reduce such friction of the machine.

Figure 7 shows the theoretical shear behaviors of NRB, LRB, and EQS isolators, where X and Y denote the lateral (shear) displacement and the lateral (shear) load applied by the actuator, respectively. In Figure 7a, representing the hysteresis of NRB, the isolator shows quasi-linear behavior. In Figure 7b, representing the shear behavior of LRB and EQS, the area enclosed in the hysteresis is larger owing to the additional damping brought by these two types of isolators. 

The shear stiffness ($K_{h}$) and equivalent damping ($h_{eq}$) can be derived by Equations (4) and (5) using the test results.
(4)$$K_{h}=\frac{Q_{2}-Q_{1}}{X_{2}-X_{1}},$$
(5)$$h_{eq}=\frac{2\cdot ∆W}{\pi K_{h}{\left(X_{2}-X_{1}\right)}^{2}},$$
where $Q_{1}$ and $Q_{2}$ are the maximum and minimum shear forces, respectively; $X_{1}$ and $X_{2}$ are the maximum and minimum displacements corresponding to $Q_{1}$ and $Q_{2}$, respectively; and ∆W is the area enclosed by the hysteresis curve and represents the energy absorbed by the isolator. 

## 3. Test Results and Discussion

### 3.1. Compressive Load Test Results

All the specimens were first loaded monotonically until the design vertical load (*p*_0_), and then cyclic compressive load was applied three times with a range between 70% (*p*_1_) and 130% (*p*_2_) of the design vertical load. Table 4 arranges the compressive test results, and Figure 8 plots the compressive force–displacement curves of NRB, LRB, and EQS. 

Noting that the elastomeric bearings have limitations in their vertical stiffness with a design vertical stiffness of 1680 kN/mm for NRB and 1995 kN/mm for LRB, Table 4 exhibits some impropriety in the vertical stiffness for LRB. Specimens #1 and #3 of LRB showed lower values than the limit value, which is 70% of the design vertical stiffness (1397 kN/mm), whereas all NRB specimens showed higher values than 70% of the design vertical stiffness (1176 kN/mm). In Figure 8, EQS showed similar vertical stiffness to the other types of isolator, but permanent displacement occurred after the peak load. This latter observation stresses the necessity to apply strict quality management on the commercial seismic isolators.

### 3.2. Compressive-Shear Test Results

As stated in the test guideline for bridge isolators, all the specimens were horizontally loaded three times within a range of design displacement. The shear force–displacement curves measured for the specimens are plotted in Figure 9. The compressive-shear test results are provided in Table 4 and can be compared with the design properties listed in Table 1 and Table 2. It appears that the shear stiffness of NRB and LRB can satisfy the design limit (90% of design shear stiffness). The shear stiffness of NRB was 25.9% higher than the design shear stiffness of 0.768 kN/mm. Even if the average shear stiffness of LRB fell below the design value, the design criterion was satisfied according to the guideline. On the contrary, the shear stiffness of EQS reached 85.1% of the design value and did not satisfy the design criterion. In view of EDC, LRB showed 33.119 kN·mm—that is, 130% of the design EDC—while EDC of EQS was 16.303 kN·mm—that is, 86% of the design value. Consequently, the considered commercial EQS failed to satisfy the test guideline for the bridge isolator despite its advantages in terms of size and maintenance. Here, this latter observation also stresses the necessity to apply strict quality management on the commercial seismic isolators.

## 4. Conclusions

A series of compressive tests and compressive-shear tests were conducted on NRB, LRB, and EQS specimens to examine the response characteristics of representative seismic isolation systems. The following conclusions are drawn:NRB showed relatively more consistent test results than LRB and EQS. LRB, which has improved energy dissipation capacity compared with NRB, exhibited lower performance than the design compressive stiffness. Though LRB and EQS have performance superior to that of NRB in terms of energy absorption capacity, only NRB satisfied the design criteria. It is, therefore, primordial to examine the quality of the isolators before their application to the structures.EQS is a novel seismic isolator with improved features. It is known that EQS can resist relatively larger vertical load than other kinds of isolators, and that it can be easily fixed after an earthquake. However, permanent deformation occurred when the design vertical load was applied in the test. Such unexpected deformation of the isolator can affect the friction between the components of EQS. Although EQS showed relatively high shear stiffness, the test results of EDC did not satisfy the target value in accordance with the test guideline for the bridge isolator. Therefore, it can be concluded that strict quality management on the commercial seismic isolators should be implemented.During the lifetime of the isolators, ageing and air temperature are important factors that affect the degradation of the rubber and friction coefficient of the sliding surface, respectively [23]. Therefore, further research will be conducted with considerations of durability.

## Figures and Tables

**Figure 1 materials-13-05247-f001:**
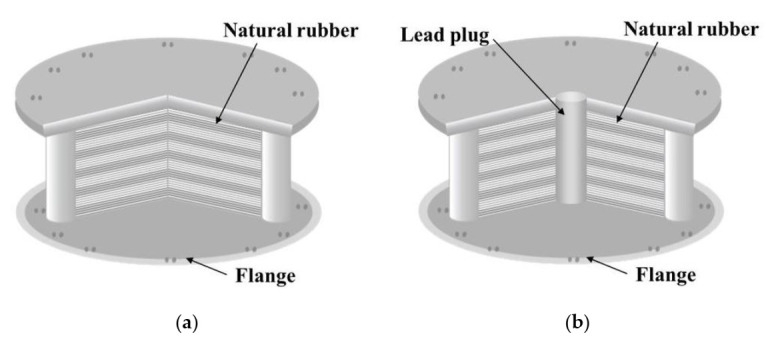
Elastomeric bearings: (**a**) Natural Rubber Bearing (NRB); (**b**) Lead Rubber Bearing (LRB).

**Figure 2 materials-13-05247-f002:**
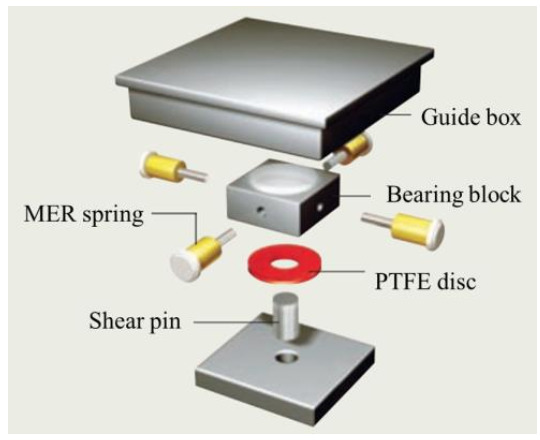
Composition of Eradi-Quake System (EQS).

**Figure 3 materials-13-05247-f003:**
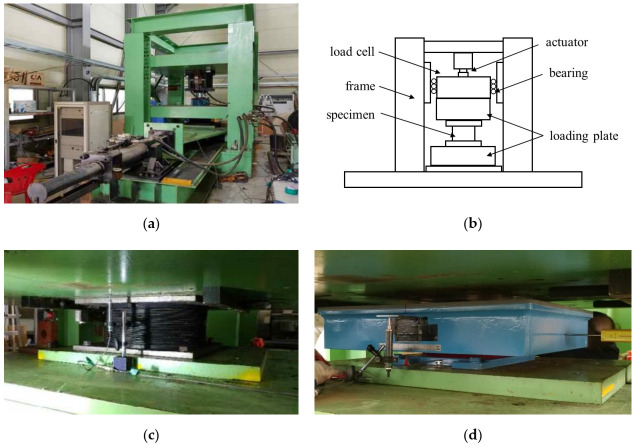
Compressive test setup: (**a**) photograph of compressive test machine; (**b**) conceptual drawing of compressive test; (**c**) installed Linear Variable Displacement Transducer (LVDT) for LRB and NRB; (**d**) installed LVDT for EQS.

**Figure 4 materials-13-05247-f004:**
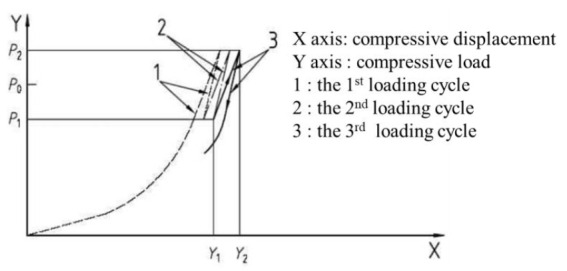
Details of biaxial loading adopted in compressive test of seismic isolators.

**Figure 5 materials-13-05247-f005:**
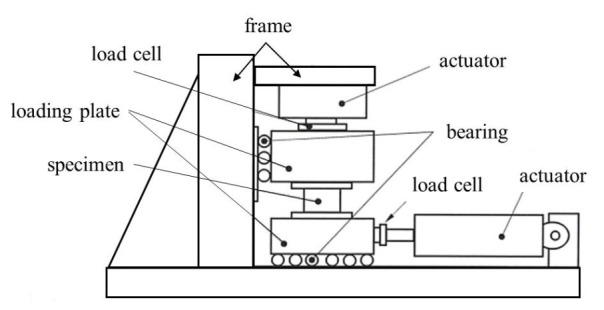
Conceptual drawing of compressive-shear test setup.

**Figure 6 materials-13-05247-f006:**
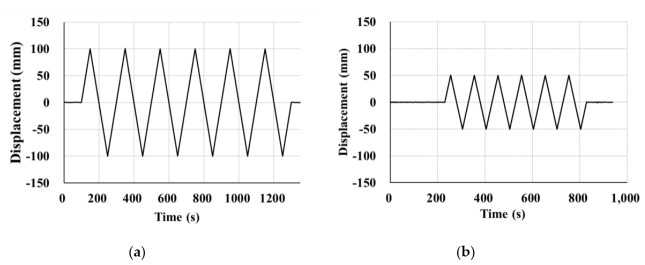
Lateral (shear) loading history: (**a**) NRB and LRB; (**b**) EQS.

**Figure 7 materials-13-05247-f007:**
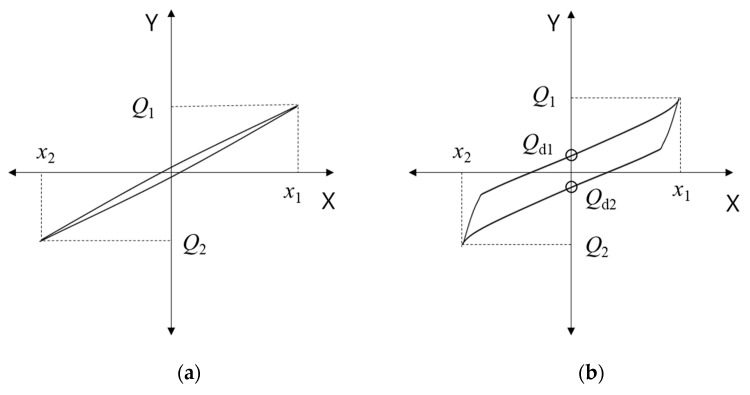
Shear behavior of isolators: (**a**) NRB; (**b**) LRB and EQS.

**Figure 8 materials-13-05247-f008:**
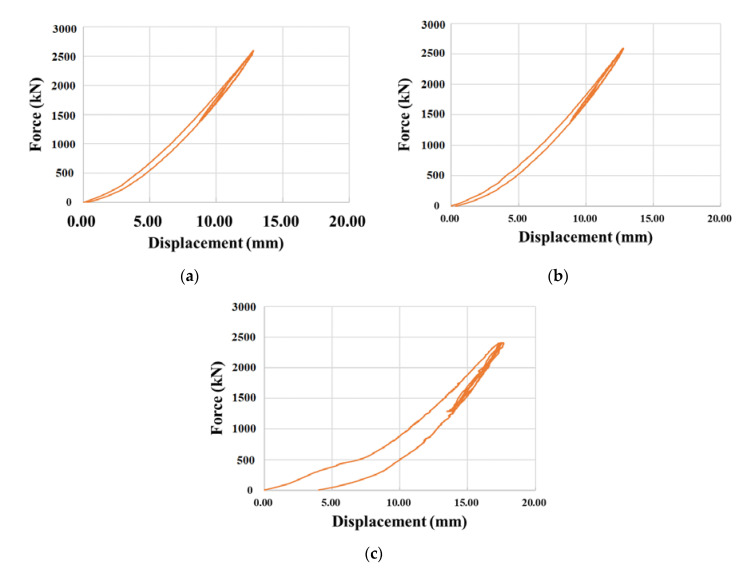
Measured compressive force–displacement curves: (**a**) NRB; (**b**) LRB; (**c**) EQS.

**Figure 9 materials-13-05247-f009:**
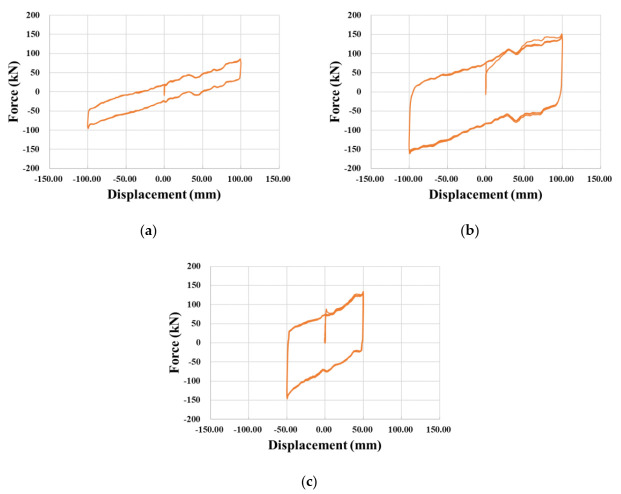
Measured shear force–displacement curves: (**a**) NRB; (**b**) LRB; (**c**) EQS.

**Table 1 materials-13-05247-t001:** Design properties of NRB and LRB specimens.

Isolator	*d* (mm)	*P_v_* (kN)	*G* (MPa)	*K_v_* (kN/mm)	*K_h_* (kN/mm)	*P_y_* (kN)	EDC (kN·mm)	*P_b_* (kN)
LRB	500	2945	0.392	2.057	1.414	67.5	25.118	12,824
NRB	500	2945	0.392	1.865	0.768	-	-	10,766

**Table 2 materials-13-05247-t002:** Design properties of EQS specimens.

Isolator	*L_dead_* (kN)	*L_live_* (kN)	*L* (kN)	*P_v_* (kN)	*d*_max_ (mm)	*f_max_*	*f_mim_*	*K_eff_* (kN/mm)	EDC (kN·mm)
EQS	1350	500	1850	2775	±50	0.138	0.055	3.219	18.900

**Table 3 materials-13-05247-t003:** Details of biaxial loading adopted in compressive test by type of seismic isolators.

Isolator	Diameter, *d* (mm)	Design Vertical Load, *p*_0_ (kN)	*p*_1_ = 70% × *p*_0_ (kN)	*p*_2_ = 130% × *p*_0_ (kN)
LRB	500	2000	1400	2600
NRB	500	2000	1400	2600
EQS	305	1850	1295	2405

**Table 4 materials-13-05247-t004:** Compressive test results of NRB, LRB, and EQS specimens.

Isolator		Compressive Stiffness, *K_v_* (kN/mm)	Shear Stiffness, *K_h_* (kN/mm)	EDC (kN·mm)	Equivalent Damping, *h_eq_*
NRB	Specimen #1	1236	0.9423	-	-
	Specimen #2	1248	0.9975	-	-
	Specimen #3	1193	0.9611	-	-
	Average	1226	0.9670	-	-
LRB	Specimen #1	1347	1.3008	32,574	0.3995
	Specimen #2	1418	1.3139	32,536	0.3951
	Specimen #3	1361	1.3069	34,247	0.4181
	Average	1375	1.3072	33,119	0.4042
EQS	Specimen #1	1370	2.7400	16,303	0.3788

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
