# Peer review of "Comparing Rubber Bearings and Eradi-Quake System for Seismic Isolation of Bridges"

_materials, 2020, doi:10.3390/ma13225247_

Round 1

Reviewer 1 Report

I have reviewed the manuscript "Comparing Properties of Rubber Bearings and Eradi Quake System" by Chang Beck Cho, Young Jin Kim, Won Jong Chin and Jin-Young Lee. It can be considered as a short communication. The paper may be of interest for both researchers and professional engineers. It needs some revisions to be published in Materials. Here are my comments.
1. Literature review is good. However, some recent works on deterioration effects of the isolation systems during their lifetime and on their impact on the seismic behaviour of the structure should be added. Useful information can be found in the following paper:
Mazza F. Effects of the long-term behaviour of isolation devices on the seismic response of base-isolated buildings. Structural Control and Health Monitoring 2019; 26(4), Article number e2331.
2. Eradi-Quake System should be better detailed (see section 1.3).
3. Friction properties of the Eradi-Quake System should be added in Table 2.
4. Loading pattern for compressive-shear tests should be added (see section 2.2.2).

Author Response

The responses to the Reviewers’ Comments are given below. The authors wish to thank the editors and reviewers for their time in effort in reviewing our manuscript. We hope the changes listed have made the manuscript suitable for publication and we look forward to your response.

Reviewer 2 Report

Materials-987638

Comparing Properties of Rubber Bearings and Eradi Quake System

This paper presents an experimental study on several seismic isolators for bridges. The research is wide and relevant, the paper is somehow well organized and not very badly written, the research approach is sound and consistent, the authors reveal a deep understanding of the analyzed issue and have carried out an extensive and costly effort (even testing); therefore, this study is useful and should be spread among the scientific and technical communities (in other words, the paper deserves publication). However, in my opinion, the paper cannot be published in its present form, mainly because poor writing. The authors are kindly encouraged to resubmit their paper after addressing the following observations.

GENERAL COMMENTS

  • Language. The text is not well written; this reviewer has experienced many difficulties in understanding the text, and significant improvement is strongly required. The use of any professional correction service (specialized in technical writing) is a must; written proof of such process will be requested in further reviews of this paper.
  • Title. The title is not considered correct as does not describe adequately the research; it needs to be improved to be more appealing and better fit to the paper content. My suggestion (not enforced): “Comparing Rubber Bearings and Eradi Quake System for Seismic Isolation of Bridges. The authors might prefer to use their own words.
  • Abstract. Due to the language issues, the description provided by the Abstract is not considered correct.
  • Keywords. The proposed keywords are considered basically correct, but “seismic isolation of bridges” should be added.
  • Acronyms. Only a moderate number of acronyms are used along the text (SUT, NRB, LRB, EQS, EDC, MER, PTFE, LVDT). Given that most of them are well established, a list of acronyms is not considered necessary, provided that all the acronyms are defined the first time they appear.
  • Organization. The overall organization of the paper is considered rather correct; however, the paper is too long and should be significantly shortened. In order to do this, more concise wording is suggested, and duplicities should be avoided (some parameters are described in the text and in Tables).
  • Apparently, this paper deals only with seismic isolation of bridges; in other words, buildings isolation is not contemplated. This should be made clear in the title, keywords, abstract, conclusions, and even in the main body of the paper.
  • In subsection 2.1, the authors claim that “seven isolators were tested”; conversely, Tables 1 and 2 describe only three devices. Please clarify.
  • The EDC presented in Tables 1 and 2, being a design parameter, should be described.
  • The mechanical parameters of rubber bearings are not uniform; conversely, they exhibit big dispersion (in the design codes, it is ordinarily accounted for by λ coefficients). The authors should comment on this (compared to EQS).
  • Figure 4 should indicate the most important elements (bearing plate, mass blocks, etc.).
  • Figure 9.c is highly unfeasible, both because of the huge maximum displacement and the permanent one. The authors should describe the origin of this plot; if it has been obtained by double integration of the acceleration, most probably long period noise has been introduced.
  • Figure 12 shows that, apparently, no sensors were installed in the rubber elements. This lack is considered surprising, and requires deep justification.
  • The provided retrofit elements are considered too voluminous; any effort leading to reducing its size would be welcome.
  • Conclusions. This section is not too long and is sufficiently self-contained (can be read without reading the rest of the paper), but is not considered adequate, mainly because of the language issues.

PARTICULAR COMMENTS (not comprehensive)

  • Please replace “brides” with “bridges” along the text.

Author Response

(The authors gave the same response as above.)

Round 2

Reviewer 2 Report

All my concerns have been addressed; the paper can be published